# Is There a Correlation between the Second-to-Four Digit Ratio (2D:4D) and Endometriosis? Results of a Case-Control Study

**DOI:** 10.3390/jcm12052040

**Published:** 2023-03-04

**Authors:** Laura Buggio, Marco Reschini, Paola Viganò, Dhouha Dridi, Giulia Galati, Alessandra Chinè, Francesca Giola, Edgardo Somigliana, Laura Benaglia

**Affiliations:** 1Gynecology Unit, Fondazione IRCCS Ca’ Granda Ospedale Maggiore Policlinico, 20122 Milan, Italy; 2Infertility Unit, Fondazione IRCCS Ca’ Granda Ospedale Maggiore Policlinico, 20122 Milan, Italy; 3Department of Obstetrics and Gynecology, Sapienza University, 00185 Rome, Italy; 4Department of Clinical Sciences and Community Health, Università degli Studi, 20122 Milan, Italy

**Keywords:** endometriosis, digit ratio, 2D:4D, intrauterine origin, deep endometriosis, endometrioma

## Abstract

The second-to-four digit ratio (2D:4D) has been proposed as a marker of prenatal hormonal exposure. It is suggested that prenatal exposure to androgens results in a shorter 2D:4D ratio, whereas a prenatal oestrogenic environment results in a longer one. In addition, previous research has shown an association between exposure to endocrine-disrupting chemicals and 2D:4D in animals and humans. On the endometriosis side, hypothetically, a longer 2D:4D ratio, reflecting a lower androgenic intrauterine milieu, could represent an indicator of the presence of the disease. In this light, we have designed a case-control study to compare 2D:4D measurements between women with and without endometriosis. Exclusion criteria included the presence of PCOS and previous trauma on the hand that could impact the measurement of the digit ratio. The 2D:4D ratio of the right hand was measured using a digital calliper. A total of 424 participants (endometriosis *n* = 212; controls *n* = 212) were recruited. The group of cases included 114 women with endometriomas and 98 patients with deep infiltrating endometriosis. The 2D:4D ratio was significantly higher in women with endometriosis compared to controls (*p* = 0.002). There is an association between a higher 2D:4D ratio and the presence of endometriosis. Our results support the hypothesis claiming potential influences of intrauterine hormonal and endocrine disruptors exposure on the onset of the disease.

## 1. Introduction

Endometriosis is a chronic, oestrogen-dependent, inflammatory disease characterised by the presence of endometrium-like epithelium and/or stroma outside the endometrium and myometrium [1,2]. Endometriosis affects about 5% of women of reproductive age [1]. The pathogenesis of the disease is still to be precisely defined. However, recently, the potential intrauterine origin of endometriosis has been gaining consensus [3,4]. An intrauterine hormonal environment characterised by an imbalance in favour of a low androgenic milieu and exposure to endocrine disruptors may be a risk factor for developing endometriosis in adult life [5,6,7,8,9,10,11,12,13]. The second-to-four digit ratio (2D:4D) has been proposed as a marker of prenatal hormonal exposure [14]. The 2D:4D ratio is a sexually dimorphic feature and represents the ratio between the length of the index finger (2D) and the length of the ring finger (4D) [15,16]. In recent decades, numerous studies have been conducted to establish whether the 2D:4D ratio is a reliable indicator of the effects of prenatal sex hormones on the body and a predictor of the development of multiple disorders [17]. This sexual dimorphism in 2D:4D ratios is apparent by two years of age and seems to be established early in life, possibly by the 14th week of gestation [18]. It is suggested that prenatal exposure to androgens results in a lower 2D:4D ratio, whereas a prenatal oestrogenic environment results in a higher one [15]. In addition, previous research showed an association between exposure to endocrine-disrupting chemicals and 2D:4D in animals and humans [19,20,21,22].

Regarding the potential association between the 2D:4D ratio and female reproductive disorders, previous studies prevalently focused their attention on polycystic ovary syndrome (PCOS). Numerous studies have evaluated this anthropometric biomarker in women with PCOS with contradictory results [22,23,24,25,26]. Interestingly, endometriosis and PCOS are both characterised by an altered function of the female hypothalamic–pituitary–gonadal axis (HPG). The function of the HPG could be influenced by different levels of prenatal androgens [27]. In line with this theory, Crespi [28] suggested that endometriosis and PCOS are expected to show a pattern of opposite causes and phenotypes due to high prenatal androgens increasing the risk of PCOS and low prenatal androgens increasing the risk of endometriosis.

On the endometriosis side, hypothetically, a longer 2D:4D ratio, reflecting a lower androgenic intrauterine milieu, could represent an indicator of the presence of the disease [28]. Only one study [25] has investigated the 2D:4D ratio in women with endometriosis without identifying any difference between cases and controls. However, the study was underpowered for definite conclusions. On the other hand, a recent Israeli study [19] identified an association between a higher 2D:4D ratio and heavier menses bleeding and dysmenorrhea.

To gain insight into the potential intrauterine origin of endometriosis, we have designed a large case-control study to compare 2D:4D measurements between women with and without the disease.

## 2. Materials and Methods

This case-control study was conducted in the Fondazione Ca’ Granda Ospedale Maggiore Policlinico of Milano, which includes a tertiary referral centre for the study and management of endometriosis and its related infertility. Participants were recruited from July 2021 to October 2022. Cases included women with a past surgical diagnosis of endometriosis or with a current nonsurgical diagnosis of the disease. Nonsurgical diagnoses were based on previously published criteria [29,30,31,32]. Women with a history of superficial endometriosis (typically diagnosed at laparoscopy) were also excluded [33]. Women with endometriosis were subcategorised into two groups: deep endometriosis (DE) and ovarian endometrioma (OMA). The DE group included women with rectovaginal plaques, bowel lesions, intrinsic ureteral endometriosis, and deep endometriosis infiltrating the pouch of Douglas and parametria. Women with both DE and OMA were included in the group of DE, as the former lesions are considered more severe than the latter ones [34]. In the same period, women attending our outpatient clinics for periodic well-woman visits, contraception, severe male infertility, and cervical cancer screening programme and without a previous clinical or surgical diagnosis of endometriosis were enrolled as the control group. Endometriosis was excluded based on gynaecological history, pelvic transvaginal ultrasound, gynaecological bimanual examination, and visual inspection of the posterior vaginal fornix. Women reporting a previous trauma on the evaluated hand that could impact the measurement of the digit ratio were excluded from both study groups. Women with PCOS, according to the 2018 definition [35], were also excluded from both groups.

In women who agreed to participate, a resident in gynaecology measured the 2D:4D digit ratio of the right hand using a digital calliper (Borletti CDJB15 150-mm Digital Calliper). All measurements were made by only three residents. They were blinded to the condition of the woman when called for the measurement. The digit lengths were measured on the right hand’s ventral surface, from the digit’s basal crease to the finger’s tip in the midline (unit of measure millimeters) (Figure 1).

The 2D:4D ratio was calculated by dividing the length of the index finger by the length of the ring finger. We decided to measure only the 2D:4D ratio on the right hand because previous studies suggested that the right hand is more sensitive to androgens [23,25]. In addition, data were collected on standardised forms, including demographic information and clinical characteristics.

The local Institutional Review Board (Comitato di Etica Milano Area B) approved the study (approval no. 980_2021bis). All participants provided written informed consent before starting the measurement of the biomarker.

The sample size (at least 150 women per group) was calculated setting type I and II errors and 0.05 and 0.20, expecting a 2D:4D ratio of 0.98 ± 0.03 in non-affected cases [25], and deeming interesting demonstrating that in women with endometriosis, the ratio could be more than 0.99. On these bases, we had to recruit about 150 women per group. However, given the relative weakness of the basic assumption used (regarding mean, SD, and distribution), we aimed for at least 200 women per group.

Data were analysed using the software Statistical Package for Social Sciences (SPSS 27.0, International Business Machines Corporation (IBM), IL, USA). All data were initially examined for normality using the Kolmogorov–Smirnov test: the normally distributed data were analysed with the Student’s t-test, while the non-normally distributed data were analysed with the Mann–Whitney test. The frequency of patients’ characteristics was compared with the Chi-square test. Data are presented as number (%), mean ± Standard Deviation (SD), and median interquartile range (IQR). *p* values below 0.05 were considered statistically significant.

## 3. Results

A total of 424 participants (endometriosis *n* = 212; controls *n* = 212) were recruited for this study. The group of cases included 114 women with OMAs and 98 patients with DE. The deep infiltrating endometriosis group comprised 50 patients with rectovaginal endometriotic plaques, 34 with deep lesions infiltrating the pouch of Douglas and parametria, 9 with full-thickness bowel lesions, and 5 with intrinsic ureteral endometriosis. In the endometriosis group, 79 participants (37%) had a surgical diagnosis of the disease. The characteristics of the 133 participants (63%) with a nonsurgical diagnosis are summarised in Table 1. One hundred-forty-eight (34.9%) women were seeking pregnancy (75 cases and 73 controls, respectively).

The general characteristics of the participants are shown in Table 2. Although the median age in the endometriosis group was 37 years, significantly higher than controls (*p* < 0.01), the other variables did not differ between the study groups. Considering that the digit length does not modify during life except for very advanced age, the statistical difference in participants’ age should not be considered a bias in the study.

The right hand 2D:4D digit ratio was not normally distributed. Therefore, we opted for non-parametric statistics. The ratio was significantly higher in women with endometriosis compared to controls (Figure 2 and Table 3) (*p* = 0.002).

The significant association remained when exclusively focussing on women with OMAs (*p* = 0.002). In contrast, the association was no more significant when the analysis was restricted to women with DE forms (*p* = 0.07). In addition, the association was still not significant when analysing women with solely DE forms (i.e., excluding those who have associated forms OMA + DE; *p* = 0.08, Table 4). Finally, we performed a sub-analysis comparing women with a surgical diagnosis of endometriosis and controls (*p* = 0.001; Table 4).

## 4. Discussion

In the present study, the 2D:4D ratio was significantly higher in the endometriosis group, particularly in women with OMAs. The association remained when exclusively focussing on women with OMAs but was lost when focussing on those with DE. We interpreted this latter finding as a type II error. The type II error could be explained by the small numerosity of the sample size and could be solved by increasing the number of women with DE.

Overall, the results of our study support the tested hypothesis (i.e.,: longer 2D:4D in the endometriosis group).

Endometriosis is a complex disease with undefined pathophysiology involving genetic factors and environmental influences [36]. Recent findings suggest that the disease may originate due to endocrine exposure during intrauterine life [37]. A low ratio of testosterone-to-estradiol during foetal life may play a crucial role in endometriosis onset and progression [4,28,37]. Of relevance here is that previous studies have already demonstrated an association between the disease and specific phenotypic characteristics, particularly in women with deep endometriosis forms [8,38,39]. For example, an association with pigmentary traits, i.e., higher numbers of naevi and freckles and the presence of blue eyes have been documented [8,38]. Furthermore, women with endometriosis had a lower body mass index (BMI) and waist-to-hip ratio (WHR) compared to women without the disease [40,41,42,43,44]. All the above findings support the potential role of oestrogen in the pathogenesis of the disease, already from the intrauterine life. Our findings also support the recent hypothesis of a group of evolutionary biologists who were persuaded that endometriosis could derive from a lower intrauterine exposure to androgens [28]. Evidence in favour of this theory is, however, not univocal. Another biomarker of intrauterine exposure to sex steroids or endocrine disruptors is represented by anogenital distance (AGD). A shorter AGD, the distance measured from the anus to the genital tubercle [10], has been linked to a more oestrogenic uterine milieu, and to a higher endometriosis risk, particularly with deep infiltrating forms [45,46,47,48]. Notably, in a previous study, we were not able to confirm this association [49].

To our knowledge, only one study [25] has previously evaluated the potential association between 2D:4D and endometriosis. Peters et al. [25] evaluated both AGD and the 2D:4D digit ratio in 172 women (endometriosis *n* = 43; Mayer–Rokitansky–Kuster-Hauser syndrome *n* = 43; PCOS *n* = 43; controls *n* = 43). The authors observed an association between a shorter AGD and the presence of endometriosis, whereas the digit ratio did not differ between the groups. The study was, however, underpowered for definite conclusions [28]. In 2020 [19], an Israelian study on 187 pregnant women revealed an association between a higher digit ratio and heavier menses bleeding and dysmenorrhea, two of the most frequently reported symptoms in endometriosis patients.

Strengths of our study include the large sample size and a direct measure of digit length, as an indirect measure may distort the 2D:4D ratio [19,50,51]. As for any case-control study, the choice of controls may represent a source of bias. In the present study, endometriosis was ruled out based on gynaecological and ultrasonographic examination. Therefore, we cannot exclude having inadvertently included some cases among controls. However, the prevalence of asymptomatic disease in the general population is modest, and misdiagnosis should be more likely for superficial peritoneal forms, a condition of uncertain clinical value [1,33]. In addition, we decided to enrol both women with a surgical and sonographic diagnosis of endometriosis. One could argue that in the latter group, a definitive diagnosis based on histological findings is lacking. However, as recently suggested by ESHRE guidelines [52], diagnostic laparoscopy should no longer be used as the first-line approach in the diagnosis of the disease. The use of imaging techniques (i.e., ultrasonography) has been repeatedly demonstrated to be highly accurate and reliable [53,54,55].

Our results provide evidence of a potential association between a biomarker of the hormonal prenatal environment in women and the presence of endometriosis. Our findings, if confirmed, could serve as predictors of the disease and have implications for endometriosis in terms of prevention and clinical practice.

In conclusion, there is an association between a higher 2D:4D ratio and the presence of endometriosis. Our results could suggest the hypothesis claiming potential influences of intrauterine hormonal and endocrine disruptors exposure on the onset of the disease. However, further evidence is needed to replicate our results and to explore further this fascinating pathogenic hypothesis.

## Figures and Tables

**Figure 1 jcm-12-02040-f001:**
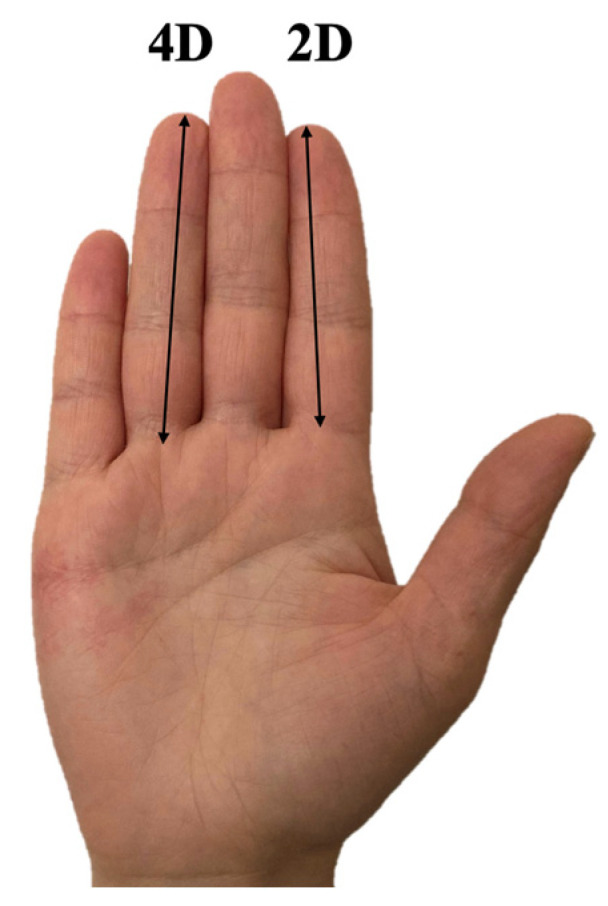
The 2D:4D ratio is calculated by dividing the length of the index finger of a given hand by the length of the ring finger of the same hand.

**Figure 2 jcm-12-02040-f002:**
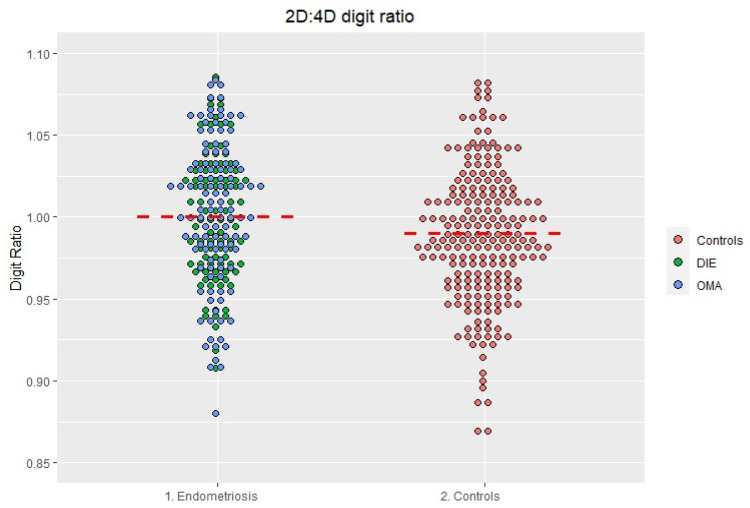
Distribution of the 2D:4D ratio in the endometriosis and in the control group. The red line represents the median.

**Table 1 jcm-12-02040-t001:** Characteristics of participants with a nonsurgical diagnosis of the disease.

Characteristics	*n*
Diagnostic method:	
- Ultrasonography	132
- Magnetic resonance imaging (MRI)	1
Type of endometriosis:	
- Ovarian endometriomas (OMA)	81
- Rectovaginal endometriosis (RV)	23
- Parametrial	4
- Ureteral endometriosis	2
- Bowel endometriosis	2
- Douglas pouch lesions	1
- Combined forms	
• OMA + RV	13
• OMA + Douglas pouch lesions	4
• OMA + Douglas + parametrial	3

OMAs: ovarian endometriomas; RV: rectovaginal endometriosis.

**Table 2 jcm-12-02040-t002:** Distribution of baseline characteristics of women with endometriosis (*n* = 212) and controls (*n* = 212).

Characteristics *	Endometriosis *n* = 212	Controls *n* = 212
Age (years)	37 [32–42]	34 [28–39]
BMI (kg/m^2^)	21.9 [20.0–24.2]	21.5 [19.5–24.2]
Ethnicity:		
Caucasian	193 (91%)	199 (94%)
African	2(1%)	5 (2%)
Asian	6(3%)	6 (3%)
Latino	11 (5%)	2 (1%)
Smoking		
Yes	39 (18%)	43 (20%)
No	158 (75%)	152 (72%)
Previous smoker	15 (7%)	17 (8%)
Age at menarche	12 [11–13]	12 [11–13]
Parity		
Nulliparous	158 (75%)	160 (75%)
Pluriparous	54 (25%)	52 (24%)
Previous Miscarriages	22 (10%)	25 (12%)
Previous IVF	78 (37%)	72 (34%)

* Data are reported as median [interquartile range] or number (percentage). BMI: body mass index.

**Table 3 jcm-12-02040-t003:** Comparison of right hand 2D:4D digit ratio of women with endometriosis and controls.

Characteristics *	Endometriosis (*n* = 212)	Controls (*n* = 212)	*p*
**Whole study groups**			
Length index finger (2D)	66.75 [63.00–70.29]	66.18 [62.92–69.19]	0.16
Length ring finger (4D)	67.00 [63.79–70.41]	66.97 [64.09–70.17]	0.81
2D:4D digit ratio	1.00 [0.97–1.03]	0.99 [0.96–1.02]	0.002
**Ovarian endometriomas (OMAs)**	*n* = 114	*n* = 212	
Length of the index finger (2D)	67.10 [63.00–70.75]	66.18 [62.92–69.19]	0.09
Length of the ring finger (4D)	67.20 [63.22–71.69]	66.97 [64.09–70.17]	0.82
2D:4D digit ratio	1.00 [0.98–1.03]	0.99 [0.96–1.02]	0.002
**Deep endometriosis (DE)**	*n* = 98	*n* = 212	
Length of the index finger (2D)	66.19 [62.93–69.44]	66.18 [62.92–69.19]	0.59
Length of the ring finger (4D)	66.77 [64.27–69.04]	66.97 [64.09–70.17]	0.51
2D:4D digit ratio	1.00 [0.97–1.03]	0.99 [0.96–1.02]	0.07

* Data are reported as median [interquartile range] or number (percentage).

**Table 4 jcm-12-02040-t004:** Comparison of right hand 2D:4D digit ratio of women with surgical diagnosis of endometriosis and controls and with different subtypes of the disease and controls.

Characteristics	Endometriosis	Controls	*p*
		(*n* = 212)	
Patients with surgical diagnosis (*n* = 79)			
Length index finger (2D)	67.05 [62.49–71.99]	66.18 [62.92–69.19]	0.20
Length ring finger (4D)	66.95 [62.83–71.39]	66.93 [64.00–70.17]	0.86
2D:4D digit ratio	1.01 [0.98–1.03]	0.99 [0.96–1.02]	0.001
OMA + OMA/DE (*n* = 173)			
Length index finger (2D)	66.85 [63.06–70.46]	66.18 [62.92–69.19]	0.10
Length ring finger (4D)	67.10 [63.76–71.00]	66.93 [64.00–70.17]	0.89
2D:4D digit ratio	1.00 [0.97–1.03]	0.99 [0.96–1.02]	0.004
DE (*n* = 39)			
Length index finger (2D)	65.91 [62.77–69.19]	66.18 [62.92–69.19]	0.98
Length ring finger (4D)	66.60 [63.97–67.71]	66.93 [64.00–70.17]	0.28
2D:4D digit ratio	1.01 [0.97–1.03]	0.99 [0.96–1.02]	0.08

OMA: ovarian endometriomas; DE: deep endometriosis.

## Data Availability

Not applicable.

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
