# Peer review of "Is There a Correlation between the Second-to-Four Digit Ratio (2D:4D) and Endometriosis? Results of a Case-Control Study"

_jcm, 2023, doi:10.3390/jcm12052040_

Round 1
Reviewer 1 Report
Buggio et al. report the results of a case-control study analyzing the potential correlation between second-to-four digit ratio (2D:4D) and endometriosis. The results are potentially significant due to the previously reported association between 2D:4D ratio and prenatal hormone exposure and the potential for prenatal hormone exposure to influence later development of endometriosis. Among 212 cases and 212 controls, a statistically significant association was found between increased 2D:4D ratio and inclusion in the endometriosis group. The large sample size is the greatest strength of the study. This study also improves upon previous studies that have used indirect measurements of 2D:4D ratio by using direct measurements with a digital caliper. However, questions about the sample groups and inclusion criteria weaken the study. Furthermore, broad conclusions about the potential intrauterine origin of endometriosis are somewhat overstated since this study examined a single putative second degree association with prenatal hormone exposures. Nevertheless, the data are of value to the research community if the patient characteristics are further clarified.
General comments:
1. A table breaking down known participant disease characteristics and method of diagnosis (as described on lines 131-138) among case groups would be helpful when considering the study population. Knowing how many cases were non-surgically diagnosed and with what specific criteria is important in evaluating the strength of the findings.
2. The authors note that the statistically significant association between 2D:4D ratio and endometriosis applied to the entire group and to the ovarian endometrioma (OMA) subgroup but not the deep infiltrating endometriosis (DIE) subgroup, which was interpreted as a type II error. The authors should discuss why this conclusion was drawn. Relatedly, since some of the women included in the DIE group also had OMA, this clouds the interpretation of the subgroup analysis. For how many patients was this the case? Would the results of the analysis be the same if the patients with both subtypes of endometriosis were grouped with OMA instead of DIE?
3. Any potential relevance of the significantly different participant age between study groups should be discussed.
4. Broad conclusions about the potential influence of intrauterine hormone and endocrine disruptor effects on endometriosis based on this study should be tempered. Since only one indirect metric (2D:4D ratio) was analyzed and did not match the recent findings from the same group looking at anogenital distance, conclusions based on the data here are very limited.
Specific comments:
1. The authors state (line 165) that the results support the tested hypothesis, but it is not completely clear what hypothesis is being evaluated in this statement. A clear and limited hypothesis statement should be made, preferably in the introduction, then be directly referenced here.
2. Please revise the section of the abstract on lines 22-23, “Case-control study.”, as this is not a complete sentence.
3. The in-line citation of reference [34] on line 95 appears to be in error.
4. Including visualization of the distribution of study subgroups (OMA and DIE) in Figure 2 would be helpful in understanding the data.
Author Response
- A table breaking down known participant disease characteristics and method of diagnosis (as described on lines 131-138) among case groups would be helpful when considering the study population. Knowing how many cases were non-surgically diagnosed and with what specific criteria is important in evaluating the strength of the findings.
We thank the reviewer for the suggestion. A new table (Table 1) has been added to describe the characteristics of the participants with non-surgical diagnosis.
- The authors note that the statistically significant association between 2D:4D ratio and endometriosis applied to the entire group and to the ovarian endometrioma (OMA) subgroup but not the deep infiltrating endometriosis (DIE) subgroup, which was interpreted as a type II error. The authors should discuss why this conclusion was drawn. Relatedly, since some of the women included in the DIE group also had OMA, this clouds the interpretation of the subgroup analysis. For how many patients was this the case? Would the results of the analysis be the same if the patients with both subtypes of endometriosis were grouped with OMA instead of DIE?
We thank the reviewer for the comment. We have added a table (Table 4) providing information on the comparison of right hand 2D:4D digit ratio of women with different subtypes of the disease (OMA + OMA/DIE; DIE alone) and controls. The type II error could be explained by the small sample size and could be solved increasing the numbers of women with DIE, this is now explained in the discussion.
- Any potential relevance of the significantly different participant age between study groups should be discussed.
Age should not impact on the digit ratio, considering that the digit length does not modify during the life except for very advanced age. For this reason, we suppose that the statistically difference in participants age should not be considered a bias in the study. We added this statement in the text.
- Broad conclusions about the potential influence of intrauterine hormone and endocrine disruptor effects on endometriosis based on this study should be tempered. Since only one indirect metric (2D:4D ratio) was analyzed and did not match the recent findings from the same group looking at anogenital distance, conclusions based on the data here are very limited.
The conclusions have been tempered according to the reviewer request.
Specific comments:
- The authors state (line 165) that the results support the tested hypothesis, but it is not completely clear what hypothesis is being evaluated in this statement. A clear and limited hypothesis statement should be made, preferably in the introduction, then be directly referenced here.
The hypothesis of our study has been already stated in the introduction: “On endometriosis side, hypothetically, a longer 2D:4D ratio, reflecting a lower androgenic intrauterine milieu, could represent an indicator of the presence of the disease”. According to the reviewer suggestion, the hypothesis has now been cleared also in the discussion.
- Please revise the section of the abstract on lines 22-23, “Case-control study.”, as this is not a complete sentence.
The phrase in the abstract has been revised
- The in-line citation of reference [34] on line 95 appears to be in error.
We thank the reviewer; the correct reference number is 35.
- Including visualization of the distribution of study subgroups (OMA and DIE) in Figure 2 would be helpful in understanding the data.
Figure 2 has been changed according to the reviewer request.
Reviewer 2 Report
Article review (Is there a correlation between the second-to-four digit ratio 2 (2D:4D) and endometriosis? Results of a case control study jcm-2124953):
Summary: This is a case control study evaluating the correlation between the second-to-four digit (2D:4D) ratio with endometriosis. The authors present data to support an association between higher 2D:4D ratio and endometriosis.
Comments/Revisions
1. Even though the accuracy of imaging modalities to diagnose endometriosis has improved, surgical evaluation is the gold standard in diagnosing this disease. Only 37% of the enrolled cases have the diagnosis of endometriosis confirmed surgically. The authors should present data on the 2D:4D ratio in the group that had surgically confirmed endometriosis compared to controls.
2. The authors suggest that a possible underlying mechanism for their observation is a more estrogenic intrauterine environment. They should add a paragraph in ‘Discussion’ to explain how this could be clinically meaningful. Could this lead to potential early screening or preventive interventions?
Recommendation
Major revision

Author Response
- Even though the accuracy of imaging modalities to diagnose endometriosis has improved, surgical evaluation is the gold standard in diagnosing this disease. Only 37% of the enrolled cases have the diagnosis of endometriosis confirmed surgically. The authors should present data on the 2D:4D ratio in the group that had surgically confirmed endometriosis compared to controls.
Data of women with a surgical diagnosis of the disease have been added in Table 4.
- The authors suggest that a possible underlying mechanism for their observation is a more estrogenic intrauterine environment. They should add a paragraph in ‘Discussion’ to explain how this could be clinically meaningful. Could this lead to potential early screening or preventive interventions?
A paragraph on the potential role of digit ratio in clinical practice has been added in the discussion section.
Round 2
Reviewer 1 Report
The authors did a thorough job addressing my questions and concerns. One additional minor correction is needed. Table 3 p values for 2D:4D digit ratio in the whole study group analysis and OMAs are 0.002, but the text indicates they are 0.003 (lines 165,180). Please correct this inconsistency.
Author Response
Thank you. We corrected the inconsistency reported.
Reviewer 2 Report
Accept
Author Response
Thank you.